# The Structural Adaptations That Mediate Disuse-Induced Atrophy of Skeletal Muscle

**DOI:** 10.3390/cells12242811

**Published:** 2023-12-10

**Authors:** Ramy K. A. Sayed, Jamie E. Hibbert, Kent W. Jorgenson, Troy A. Hornberger

**Affiliations:** 1Department of Comparative Biosciences, University of Wisconsin—Madison, Madison, WI 53706, USA; rsayed2@wisc.edu (R.K.A.S.); jhibbert@wisc.edu (J.E.H.); kwjorgenson@wisc.edu (K.W.J.); 2School of Veterinary Medicine, University of Wisconsin—Madison, Madison, WI 53706, USA; 3Department of Anatomy and Embryology, Faculty of Veterinary Medicine, Sohag University, Sohag 82524, Egypt

**Keywords:** disuse, fascicle, hypoplasia, longitudinal atrophy, muscle fibers, myofibril, myofilaments, radial atrophy, sarcomere

## Abstract

The maintenance of skeletal muscle mass plays a fundamental role in health and issues associated with quality of life. Mechanical signals are one of the most potent regulators of muscle mass, with a decrease in mechanical loading leading to a decrease in muscle mass. This concept has been supported by a plethora of human- and animal-based studies over the past 100 years and has resulted in the commonly used term of ‘disuse atrophy’. These same studies have also provided a great deal of insight into the structural adaptations that mediate disuse-induced atrophy. For instance, disuse results in radial atrophy of fascicles, and this is driven, at least in part, by radial atrophy of the muscle fibers. However, the ultrastructural adaptations that mediate these changes remain far from defined. Indeed, even the most basic questions, such as whether the radial atrophy of muscle fibers is driven by the radial atrophy of myofibrils and/or myofibril hypoplasia, have yet to be answered. In this review, we thoroughly summarize what is known about the macroscopic, microscopic, and ultrastructural adaptations that mediated disuse-induced atrophy and highlight some of the major gaps in knowledge that need to be filled.

## 1. Introduction

Skeletal muscle is the largest organ in the human body and, on average, contributes to 35% of an individual’s body mass [1]. Skeletal muscles are not only the motors that drive basic movements, but also serve numerous physiological functions (e.g., regulation of body temperature and metabolism) and can function as endocrine organs via the secretion of myokines [2,3,4,5]. Based on these points, it is not surprising that the maintenance of muscle mass has been widely implicated in health and issues associated with quality of life. For instance, low muscle mass has been correlated with an increased risk of type II diabetes, heart disease, and death [6,7,8,9,10,11], and the maintenance of muscle mass becomes particularly important during aging as the age-associated loss of muscle has been strongly linked to disability, lack of independence, mortality, and an estimated $20–40 billion in annual healthcare costs in the United States alone [12,13,14,15]. Thus, the development of therapies that can maintain, restore, or even enhance muscle mass is of great clinical and fiscal significance. However, to develop such therapies, we will first need to develop a deeper understanding of the mechanisms that regulate the size of this vital organ. 

It is well established that skeletal muscle mass can be regulated via a variety of different stimuli, and one of the most potent stimuli is mechanical loading [16,17]. Indeed, even in the 5th century BC, Hippocrates appreciated that ‘exercise strengthens and inactivity wastes’ [18]. Thus, the notion that a loss in mechanical loading can lead to a loss of muscle mass appears to have been recognized for at least 2500 years [18]. Over the past 100 years, a plethora of human- and animal-based studies have provided further support for this concept and resulted in the commonly used term ‘disuse atrophy’. These same studies have also provided insight into the macroscopic and microscopic changes that contribute to the loss of muscle mass (i.e., atrophy). Surprisingly, however, the ultrastructural adaptations that drive these changes remain far from defined. In this review, we will summarize what is currently known about the structural adaptations that mediate disuse-induced atrophy and highlight the critical gaps in knowledge that need to be filled.

## 2. Disuse Atrophy and the Models That Can Be Used to Study It

Before discussing the models that can be used to study disuse atrophy, it is necessary to have a precise definition for the term ‘disuse’. In this review, disuse will be defined as any condition in which there is a maintenance or reduction in the mean passive tension along with a reduction in the mean active tension generated by the muscle. This definition is important to keep in mind because it will exclude certain skeletal muscles that are often examined in commonly used models of disuse atrophy. Furthermore, it is known that a variety of diseases such as congestive heart failure, cirrhosis, and cancer can lead to disuse [19,20,21]. However, such diseases often present with an increase in circulating atrophic factors such as cytokines and cortisol [21,22,23,24]. The structural adaptations that contribute to disuse atrophy in diseased versus otherwise healthy individuals may be different and, therefore, this review will only focus on the structural adaptations that mediate disuse atrophy in subjects that are free of any apparent disease. 

### 2.1. Space Flight

Skeletal muscle atrophy is a widely recognized consequence of prolonged spaceflight, and the disuse that results from the microgravity environment of space is considered to be the key driver of the atrophic response [25,26,27]. However, as recently summarized by Lee et al., 2022, additional factors such as radiation and the stresses associated with living in space might also play a role [27]. Nonetheless, the most prominent atrophic response occurs in antigravity muscles (e.g., the soleus) which suggests that disuse is the primary driver [27,28]. Importantly, as defined in this review, some muscles of the body would not qualify as being subjected to disuse during space flight. For instance, humans assume a plantarflexed foot drop posture during spaceflight, and in this position, the plantarflexors are maintained in a shortened position whereas the dorsiflexors are maintained in a lengthened position [29,30]. The lengthening of the dorsiflexors would result in a chronic increase in the passive tension that is generated by the dorsiflexor muscles, and thus, these muscles would not fulfill the definition of being subjected to disuse. While this may seem like a trivial point, a chronic increase in length/passive tension can profoundly influence the structural adaptions that occur when a muscle is subjected to a chronic decrease in active force generation [31,32,33,34], and it is our hope that the reader will gain a deep appreciation for this concept as they work through the remainder of the review. 

### 2.2. Bed Rest

In the 19th century, bedrest was advocated as the primary line of treatment for a variety of ailments [35]. However, shortly after World War II, classic studies in humans, such as the one performed by Dietrick and Whedon (1948), began to expose the negative impact that prolonged bedrest can have on various metabolic and physiological parameters (e.g., skeletal muscle mass and strength) [36]. Since that time, it has become clear that the disuse that occurs during prolonged bed rest can lead to profound atrophy, particularly of the primary weight-bearing muscles of the human body such as the soleus and quadriceps [37,38,39,40]. 

### 2.3. Hindlimb Suspension

Hindlimb suspension (also known as hindlimb unloading) is an animal model that has been extensively used in the study of disuse atrophy. The model was originally developed as a complement to the 6° head-down bed rest model of spaceflight in humans, and similar to space flight, the hindlimbs of the suspended animals assume a plantarflexed position [41,42,43,44,45]. As such, the dorsiflexors (e.g., tibialis anterior and extensor digitorum longus) experience a chronic increase in passive tension and therefore do not fulfill the definition of being subjected to disuse. Again, this may seem like a trivial point, but it should not be overlooked. For instance, previous studies have shown that the soleus is particularly prone to hindlimb suspension-induced atrophy, yet the atrophic response can be prevented if the soleus is maintained in a lengthened position [46,47]. Thus, we would like to reiterate that, as defined in this review, disuse requires a maintenance or reduction in the mean passive tension as well as a reduction in the mean active tension generated by the muscle. 

### 2.4. Unilateral Limb Suspension 

The unilateral limb suspension model is a human model that can be used to study the impact that disuse has on muscles that are maintained in a relatively relaxed/neutral position. Specifically, in the unilateral limb suspension model, participants wear a shoe with a 100-mm-thick sole on the weight-bearing leg and then use crutches when they need to move around. With this approach, the unloaded limb can swing freely around the hip and numerous studies have shown that it induces atrophy of the knee extensors and the plantar flexors [48,49].

### 2.5. Immobilization

Immobilization is often used in the treatment of traumatic injuries such as fractured bones and torn ligaments. It is well recognized that immobilization can lead to disuse atrophy, and a variety of human and animal models of immobilization have been developed to study the mechanisms of this phenomenon. In humans, the most commonly employed approach involves the use of a cast or brace to immobilize the ankle or knee in a relatively neutral position [50,51]. More extreme variations have been used in animals in which splints, casts, or staples have been used to immobilize the lower hindlimb in a fully plantarflexed, neutral, or fully dorsiflexed position [52,53,54]. However, as defined in this review, only muscles that are immobilized at resting/neutral or shorter than resting length would qualify as being subjected to disuse. Again, the importance of the point was highlighted many years ago when it was shown that the length at which a muscle is immobilized can determine whether it atrophies. Specifically, in 1944, Thomsen and Luco reported that when the soleus or tibialis anterior muscles of cats were immobilized at a shorter-than-normal length the muscles atrophied, whereas immobilization of the same muscles at a longer-than-normal length led to an increase in muscle mass [55]. Thus, unless otherwise stated, any discussions in this review that refer to immobilization will imply that the muscles being discussed were immobilized at resting, or shorter than resting, length. 

### 2.6. Denervation

Denervation can be caused by a variety of different conditions (e.g., motor neuron diseases such as ALS, physical transection of the nervous input, etc.), and various forms of denervation have been used as models of disuse atrophy. It is clear that denervation can lead to a dramatic atrophic response; however, numerous studies have also shown that denervation can lead to signs of muscle fiber death and a concomitant regenerative response [56,57,58,59]. These types of events are rarely, if ever, observed in the other models of disuse described above. Thus, the atrophy that occurs in response to denervation is likely the result of more than just disuse. Accordingly, studies that have employed denervation as a model of disuse will not be further considered in this review.

## 3. Overview of Skeletal Muscle Structure 

Before we discuss what is known about the structural adaptations that mediate disuse atrophy, we first want to ensure that the reader appreciates the basic structural design of skeletal muscle. In our opinion, the easiest way to appreciate this design is to consider skeletal muscle as a hierarchy of structural elements that is visible at the macroscopic level (viewable without magnification), followed by the microscopic level (viewable with standard microscopy), and finally the ultrastructural level (viewable with high-resolution microscopy). Below we will describe the structural elements that are found at each of these levels. For a more detailed description of these components, please refer to the following reviews [60,61,62,63].

Beginning with the macroscopic level, one will find that skeletal muscles are attached to bones via tendons (Figure 1A). Moving away from the bony connection, the tendon transitions into the musculotendinous junction and then into the epimysium, which is a layer of connective tissue that encases the whole muscle [60]. Underneath the epimysium are smaller organizational units known as fascicles. Each fascicle consists of a bundle of muscle fibers that is encased by a sheath of connective tissue called the perimysium [64,65]. It bears noting that in the majority of skeletal muscles, fascicles are not arranged in parallel with the long axis of the muscle, but rather run along an angle known as the pennation angle, which plays an important role in muscle force production and size [63].

At the microscopic level, a cross-section of skeletal muscle will reveal the presence of individual muscle fibers (Figure 1A–C). Each muscle fiber is encased by a membrane called the sarcolemma that, in turn, is surrounded by a sheath of connective tissue called the endomysium. Running between the muscle fibers are a variety of different types of interstitial cells including fibro-adipogenic/mesenchymal progenitors, fibroblasts, immune cells, and pericytes [66,67,68]. There is also a unique class of cells that reside between the endomysium and the sarcolemma, called satellite cells, which function as muscle stem cells and are critically important for myogenesis (Figure 1A,C) [67,68,69]. 

At the ultrastructural level, one will find that ~80% of each muscle fiber is filled with structures called myofibrils [70,71]. Myofibrils contain the basic contractile machinery of the muscle (i.e., the myosin (thick) and actin (thin) myofilaments) and are surrounded by mitochondria and a mesh-like membrane layer called the sarcoplasmic reticulum (Figure 1A,D,E). Within the myofibril, the thick and thin myofilaments are arranged in a repetitive series of structures called sarcomeres (Figure 1A). The length of sarcomeres is highly conserved within a given species, typically measuring 2.0–2.5 µm in length [72,73], and this length is determined by the distance between the two Z-discs of the sarcomere. Within the sarcomere, the thin myofilaments attach to the Z-disc and extend toward the middle of the sarcomere (M-line), where thick myofilaments attach. The area closest to the Z-discs that are occupied solely by the thin myofilaments is known as the I-band, whereas the middle portion of the sarcomere, which is occupied by the thick myofilaments, is known as the A-band (Figure 1A) [60,74,75]. Within the A-band, there is a region called the H-zone, which only contains thick myofilaments. Throughout the remainder of the A-band, there is overlap between the thick and thin myofilaments, and this overlap endows the sarcomere with the ability to actively produce force as described by the sliding filament theory of muscle contraction [76,77]. 

## 4. Disuse-Induced Atrophy at the Macroscopic Level

### 4.1. Whole Muscle

At the whole muscle level, disuse-induced atrophy can be driven by a decrease in the length and/or a decrease in the diameter/cross-sectional area (CSA) of the muscle. Atrophy resulting from a decrease in the length is known as longitudinal atrophy, whereas atrophy that results from a decrease in the CSA is referred to as radial atrophy. As described in the overview of skeletal muscle structure, muscles are attached to bones, and in adults, the length of bones is fixed [78,79,80]. Accordingly, the resting length of muscles is generally viewed as being invariable and therefore has seldom been measured in studies of disuse atrophy. Nonetheless, a handful of studies have consistently shown that when a muscle is immobilized in a shortened position, it can undergo substantial longitudinal atrophy [81,82,83,84]. For instance, Coutinho et al. (2004) reported that the length of the soleus muscle in rats decreased by 19% after 3 weeks of being immobilized in a shortened position [84], and the magnitude of this effect increased to 27% after 6 weeks of immobilization [82]. 

Although a few studies have examined the length of muscles that were subjected to disuse while being maintained in a shortened position, we are only aware of one study that has reported on the length of muscles that were subjected to disuse while being maintained in a neutral/resting position. In this case, Spector et al. (1982) immobilized the hindlimb of rats in a neutral position for 4 weeks and found that several muscles including the soleus, medial gastrocnemius, and tibialis anterior experienced a 41–49% decrease in mass, but none of these muscles revealed a significant decrease in length [81]. Importantly, in the same study, additional experiments were performed in which the aforementioned muscles were immobilized in a shortened position and, in this case, all of the muscles underwent a significant 12–21% decrease in length [81]. Thus, according to the limited data in the field, it appears that when disuse occurs in conjunction with a chronic shortening of the muscle it will lead to longitudinal atrophy at the whole muscle level, but longitudinal atrophy at the whole muscle level will not transpire when muscles are subjected to disuse in a chronically neutral position. 

The work of Spector et al. (1982) illustrates that a very large disuse-induced decrease in muscle mass can occur in the absence of longitudinal atrophy [81]. Like all materials, the mass of a given muscle will be a function of its volume and density, and we are not aware of any studies that have reported significant disuse-induced changes in muscle density. Thus, the decreases in muscle mass reported by Spector et al. were likely mediated via a decrease in volume. In skeletal muscle, volume is a function of the whole muscle CSA and the whole muscle length. Given that Spector et al. (1982) were able to observe decreases in mass in the absence of changes in length, it can be argued that the decreases in mass were driven by a decrease in the whole muscle CSA (i.e., radial atrophy) [81]. Indeed, radial atrophy at the whole muscle level has been observed in all of the models of disuse atrophy described in Section 2 including space flight [85,86], bed rest [87,88,89,90,91,92,93,94,95,96,97], unilateral lower limb suspension [98,99,100], and immobilization [81,101,102,103,104,105,106,107]. Of note, most of these studies obtained measurements of the whole muscle CSA at the midpoint along the length of the muscle (i.e., at 50% of the whole muscle length) [85,89,99]. However, disuse-induced changes in whole-muscle CSA are not always homogenous along the length of the muscle [92,95,96,97]. For instance, Franchi et al. (2022) reported that after just 10 days of bed rest in humans, a 5.8% and 7.0% reduction in the CSA of the biceps femoris short head could be detected at 30% and 50% of the muscle length (distal to proximal), respectively. Yet, at 70% of the muscle length, no significant change in CSA was observed [97]. Likewise, the same study also reported an 8.5% decrease in the CSA of the semitendinosus muscle at 30% of the muscle length, with no significant changes being observed at 50% or 70% of the muscle length. Hence, when taken together, it appears that all models of disuse can induce radial atrophy at the whole muscle level, and the magnitude of the radial atrophy can vary along the length of the muscle. 

### 4.2. Muscle Fascicles

As a basic guideline, it is helpful to consider the fact that disuse-induced changes that occur at one level of muscle structure will be mediated via the changes that occur at the preceding level within the hierarchy of the structural elements (e.g., myofilaments → myofibrils → fibers → fascicles → whole muscle). According to this guideline, the longitudinal and radial atrophy that occurs at the whole muscle level will be mediated via the changes that occur at the level of the fascicles, and when evaluating how changes at the level of the fascicles could influence the whole muscle structure, the architectural properties of the muscle must be kept in mind. The importance of this point cannot be understated because it would be very counterintuitive to propose that robust longitudinal atrophy at the whole muscle level could be mediated exclusively via radial atrophy of the fascicles, or that robust radial atrophy at the whole muscle level could be mediated exclusively via longitudinal atrophy of the fascicles. However, as illustrated in Figure 2, these are both very realistic possibilities.

Readers who are not familiar with the geometric principles described in Figure 2 are strongly encouraged to view the work of Maxwell et al. (1974) because further consideration of this work will reveal that a decrease in whole muscle length and/or whole muscle diameter/CSA could be mediated via radial atrophy of the fascicles and/or longitudinal atrophy of the fascicles, and the specific mechanism(s) involved will depend on the inherent architectural properties of the muscle as well as the physical constraints under which the adaptations occur [108]. For instance, the examples shown in Figure 2A vs. Figure 2C illustrate that when whole muscle length is considered as a fixed variable, a 25% decrease in fascicle length could lead to a 34% decrease in the whole muscle diameter but no change in the whole muscle length or the pennation angle of the fascicles. Alternatively, when whole muscle length is not considered as a fixed variable, a 25% decrease in fascicle length could lead to a 14% decrease in whole muscle length, a 20% decrease in whole muscle diameter, and a 16% increase in the pennation angle of the fascicles. From a geometric perspective, there is an endless number of possibilities, and thus the key point here is worth repeating: a decrease in whole muscle length and/or whole muscle diameter/CSA could be mediated via radial atrophy of the fascicles and/or longitudinal atrophy of the fascicles, and the specific mechanism(s) involved will depend on the inherent architectural properties of the muscle as well as the physical constraints under which the adaptations occur.

Having described how longitudinal atrophy of fascicles could induce changes in whole muscle length and/or whole muscle CSA, we will now address whether this type of atrophy occurs in response to disuse. Surprisingly, we could not find any animal-based studies that have examined the effects of disuse on fascicle length. However, numerous studies in humans have reported such measurements and have shown that longitudinal atrophy of the fascicles occurs in response to various forms of disuse including space flight, bed rest, and unilateral lower limb suspension [97,99,109,110,111,112,113,114,115]. Importantly, although it appears that longitudinal atrophy at the whole muscle level only transpires when disuse occurs in conjunction with a chronic shortening of the muscle, longitudinal atrophy at the level of the fascicles can be observed when muscles are subjected to disuse in a predominantly neutral position [99,113,114]. For instance, unilateral lower limb suspension maintains the knee and ankle in a natural/neutral position, and work by De Boer et al. (2007) revealed that the length of fascicles in the vastus lateralis muscle decreased by 8% after 23 days of unilateral lower limb suspension [99]. Likewise, Campbell et al. (2013), reported that 23 days of unilateral lower limb suspension led to a 6–11% decrease in fascicle length in all four heads of the quadriceps [113]. 

Longitudinal atrophy at the level of the fascicles has also been reported to occur with bed rest [97,109,110,112]. As a case in point, Reeves et al. (2002) reported that 90 days of bed rest led to a 10% decrease in the fascicle length of the gastrocnemius [109]. However, the length at which various muscles are maintained during bedrest has not been described. During typical sleep postures, the ankle will assume a plantarflexed position while the position of the knee can be highly variable [116,117,118,119]. If these positions are dominant during the waking portion of bedrest, then it could be assumed that the plantarflexors are maintained in a shortened position whereas the position of the knee flexors and extensors would be unpredictable. With this point in mind, one is faced with the question of whether the bedrest-induced decrease in fascicle length reported by Reeves et al. (2002) was dependent on a chronic shortening of the muscle [109]. To date, no studies have directly addressed this question; however, De Boer et al. (2008) reported a 6% decrease in the fascicle length of the vastus lateralis muscle after five weeks of bed rest [110], and Sarto et al. (2021) reported a 2.8% decrease in fascicle length of the biceps femoris after just 10 days of bed rest [112]. The vastus lateralis and biceps femoris exert opposing actions on the position of the knee, and thus, if one muscle is maintained in a shortened position then the other would have to be maintained in the lengthened position. Yet, both muscles revealed a bedrest-induced decrease in fascicle length. Thus, the available evidence indicates that disuse leads to the longitudinal atrophy of fascicles, this phenomenon does not appear to be dependent on a chronic shortening of the muscle, and according to Maxwell’s geometric model, the longitudinal atrophy of fascicles could lead to changes in whole muscle length and/or whole muscle CSA. 

Just like longitudinal atrophy, radial atrophy of fascicles could also induce changes in whole muscle length and/or whole muscle CSA (Figure 2), but whether disuse leads to radial atrophy of fascicles has not been reported. Nonetheless, a multitude of studies have examined changes in the pennation angle of fascicles and, as illustrated by Maxwell et al. (1974), there is a direct relationship between fascicle diameter and the pennation angle of the fascicles [99,108,110,113,114,115]. Specifically, Maxwell’s geometric model indicates that when muscle length is a fixed variable, then a decrease in fascicle diameter will lead to a decrease in the pennation angle of the fascicles and vice versa (Figure 2E). As described in Section 4.1, it is generally thought that changes in whole muscle length do not occur when muscles are subjected to disuse in a neutral position. Thus, if a decrease in pennation angle is observed in muscles subjected to disuse in a neutral position, then it can be inferred that the decrease in pennation angle was due to a decrease in the diameter of the fascicles (i.e., radial atrophy of the fascicles). Fortunately, several studies have reported such an effect [99,113,114]. For example, Campbell et al. (2013) reported that, after 23 days of unilateral lower limb suspension, the pennation angle of the fascicles in all four heads of the quadriceps decreased by 7–10% [113]. Moreover, the work of De Boer et al. (2007) revealed that the pennation angle of the fascicles in the vastus lateralis muscle decreased by 7.6% after 23 days of unilateral lower limb suspension [99].

In addition to the aforementioned studies, a decrease in the pennation angle of fascicles has also been reported when the average position of the muscle during disuse was less clear (e.g., during bed rest) [109,110,111,115]. As mentioned above, the plantarflexor muscles are likely maintained in a shortened position during bed rest and therefore the length of the plantarflexor muscles would be expected to either slightly decrease or remain unchanged. According to Maxwell’s geometric model, if the length of the muscle did not change during bed rest, then the decrease in the pennation angle of the fascicles would be directly mediated via a decrease in fascicle diameter. Importantly, however, if the length of the muscle decreased, then a decrease in the pennation angle of the fascicles would likely underrepresent the actual decrease in fascicle diameter that occurred [108]. Thus, although no studies have directly addressed whether disuse leads to radial atrophy of fascicles, inferences can be made, and such inferences consistently indicate that disuse leads to the radial atrophy of fascicles. 

## 5. Disuse-Induced Atrophy at the Microscopic Level

As mentioned above, disuse-induced changes that occur at one level of muscle structure reflect the changes that occur at the preceding level within the hierarchy of the structural elements. Thus, having established that disuse atrophy can lead to longitudinal and/or radial atrophy of the fascicles, we will now describe how these alterations are driven by microscopic changes that occur at the level of the muscle fibers.

### 5.1. Longitudinal Atrophy of Fascicles

Fascicles consist of a bundle of muscle fibers, and, in general, the muscle fibers run the entire length of the fascicles [120,121]. For such fascicles, longitudinal atrophy would most likely be mediated via a decrease in the length of the individual fibers. However, in some muscles, fascicles are composed of muscle fibers that only run a fraction of the fascicle length. In such fascicles, the muscle fibers possess intrafascicular terminations which consist of tapered ends that interdigitate with the tapered ends of other fibers within the fascicle [122,123,124,125]. In theory, longitudinal atrophy of fascicles with intrafascicular terminations could be mediated via a decrease in the length of the individual muscle fibers and/or the loss of muscle fibers in series. However, we are not aware of any studies that have addressed whether disuse-induced atrophy can lead to the loss of muscle fibers in series, and therefore, this section will focus on whether disuse leads to a decrease in muscle fiber length. 

Despite an exhaustive search, we could not find any human-based studies that have directly investigated the effects of disuse on muscle fiber length. Nevertheless, numerous animal-based studies have shown that when muscles are subjected to disuse in a shortened position, it leads to a decrease in fiber length [81,82,84,126,127,128,129]. For instance, Widrick et al. (2008) reported an 11% decrease in the length of soleus muscle fibers following 15 days of hindlimb unloading [128]. Likewise, immobilization of the soleus and medial gastrocnemius muscles in a shortened position for four weeks led to 14% and 26% decreases in the length of the muscle fibers, respectively [81]. Similarly, Witzmann et al. (1982) found that immobilization of the soleus in a shortened position for 42 days led to a 27% decrease in fiber length [82]. 

Although a consistent body of literature indicates that disuse in a shortened position leads to a decrease in fiber length (i.e., longitudinal atrophy of the fibers), whether the same effect happens when muscles are subjected to disuse in a neutral position is not clear. Specifically, we are only aware of one study that has directly addressed this question, and in this study, Spector et al. (1982) concluded that the length of the fibers in the soleus, medial gastrocnemius, and tibialis anterior muscles of rats was not significantly altered after 4 weeks of immobilization in a neutral position [81]. However, it bears noting that the rats in this study were quite young (namely, 200–240-g males, age not specified but estimated to be 6–7 weeks) [130]. This is noteworthy because the longitudinal growth of bones in male rats continues until approximately 20 weeks of age [131]. Thus, in the work by Spector et al. (1982), the limb of the rats may have been immobilized with the muscles in a neutral position, but the muscles were likely still experiencing a chronic lengthening stimulus from the longitudinal growth of the bones. As such, the results from Spector et al. (1982) should be interpreted with caution. Continuing with this point in mind, it is important to acknowledge the studies from the previous section which indicated that longitudinal atrophy of the fascicles occurs when muscles are subjected to disuse in a neutral position [99,113,114]. Specifically, in these studies, a decrease in fascicle length was observed in several muscles, including all four heads of the quadriceps and the lateral head of the gastrocnemius. Available evidence indicates that, in all of these muscles, the fibers run the entire length of the fascicles (i.e., there are no intrafascicular terminations) [64,132,133]. Thus, it would seem fair to argue that the disuse-induced decrease in fascicle length was mediated via longitudinal atrophy of the fibers. However, we recognize that this argument is based purely on speculation, and the only responsible conclusion that can be drawn from this section is that additional studies are needed to determine whether disuse in a neutral position leads to longitudinal atrophy of the muscle fibers.

### 5.2. Radial Atrophy of Fascicles

At the microscopic level, there are two primary mechanisms via which radial atrophy of fascicles could be mediated: (i) a decrease in the CSA of the existing muscle fibers (i.e., radial atrophy of the muscle fibers, Figure 3A), and/or (ii) a decrease in the number of muscle fibers per cross-section (i.e., muscle fiber hypoplasia, Figure 3B). These mechanisms have been investigated in the context of disuse-induced atrophy, and the available literature on these topics will be summarized in the following sections. However, before doing so, it should be noted that, in fascicles with intrafascicular terminations, radial atrophy could also result from the longitudinal atrophy of muscle fibers (Figure 3C). Despite this possibility, we are not aware of any studies that have addressed it, and therefore, this potential mechanism will not be further discussed.

#### 5.2.1. Radial Atrophy of Muscle Fibers

Radial atrophy of muscle fibers is defined as a decrease in the CSA of the muscle fibers. Having illustrated how radial atrophy of the muscle fibers can lead to radial atrophy of fascicles, we will now consider the studies that have investigated whether it occurs in response to disuse. Fortunately, this subject has been extensively studied in all models of disuse atrophy described in Section 2, including space flight [134,135,136,137,138], bed rest [139,140,141], unilateral lower limb suspension [98,113,142], hindlimb unloading [47,128,143,144,145,146,147], and immobilization [84,106,148,149,150], and the results of these studies consistently demonstrate that disuse leads to radial atrophy of the muscle fibers. For instance, Arentson-Lantz et al. (2016) reported a 24% decrease in the CSA of human vastus lateralis muscle fibers after 14 days of bed rest [141]. Likewise, mice flown aboard the space shuttle for 13 days showed a 39% decrease in the CSA of the gastrocnemius muscle fibers [138]. In addition, Desplanches et al. (1990) reported that five weeks of hindlimb suspension led to a 75% decrease in the CSA of the soleus muscle fibers in rats [143]. Thus, when taken together, it appears that radial atrophy of fascicles could be mediated, at least in part, via radial atrophy of the muscle fibers. 

#### 5.2.2. Changes in the Number of Muscle Fibers per Cross-Section

Hypoplasia refers to the loss of muscle fibers, and as illustrated in Figure 3B, hypoplasia could promote radial atrophy of the muscle fascicles. However, there are conflicting reports with regard to whether disuse induces hypoplasia, and this conflict can be attributed to the different types of methods that have been used to address the question. Specifically, two primary methods have been employed: (i) counting the total number of muscle fibers per mid-belly cross-section of the muscle [151,152,153], or (ii) digestion of the muscle’s connective tissue followed by direct counting of the total number of muscle fibers per muscle [148,154]. The direct counting method should be the most accurate, but the use of this method requires the manual dissociation and counting of thousands of muscle fibers per muscle. Consequently, most studies simply count the number of muscle fibers per mid-belly cross-section and assume that this is reflective of the total fiber number. However, it is critical to appreciate that changes in the architectural properties of the muscle (e.g., fiber length, fiber CSA, etc.) can dramatically alter the number of muscle fibers that appear in a mid-belly cross-section. Therefore, to help the reader appreciate this, we developed to-scale replicas of the soleus and plantaris muscles of the rat (Figure 4). As detailed in the figure legend, the replicas are based on architectural properties that were reported in previous studies [84,127,128,143,155,156,157], and with these replicas, we used Maxwell’s geometric model to examine how a 16% decrease in fiber length [81,82,84,127,128] and/or a 61% decrease in fiber CSA [84,127,128,143] would affect the number of muscle fibers that appear in a mid-belly cross-section. The first thing for the reader to note in this figure is that, after zooming in on panel 4B, it should be evident that all of the fibers within the soleus muscle pass through the mid-belly of the muscle (hatched box). The reason for this is that, in the soleus, [fiber length × cosine θ] is greater than one-half of the muscle length. Importantly, in muscles that exceed this critical threshold value, changes in fiber length and/or fiber CSA will not affect the number of fibers that appear in a mid-belly cross-section (Figure 4C,D). Nonetheless, in many muscles (e.g., the plantaris), [fiber length × cosine θ] does not exceed one-half of the muscle length. When this critical threshold is not exceeded, all of the fibers in the muscle will not pass through the mid-belly and, as illustrated in Figure 4B–D, changes in fiber length and/or fiber CSA of such muscles can dramatically alter the number of fibers that appear in a mid-belly cross-section. For instance, in the rat plantaris, Maxwell’s geometric model predicts that a 16% decrease in fiber length would lead to a 41% decrease in the number of fibers in a mid-belly cross-section. Thus, without an appreciation of the architectural properties of the muscle, one might mistakenly conclude the decrease in the number of fibers per cross-section was indicative of hypoplasia when, in reality, it was exclusively mediated via longitudinal atrophy of the pre-existing fibers. 

Having illustrated how the number of fibers per cross-section can be influenced by the architectural properties of the muscle, we will now consider the studies that have investigated whether disuse induces hypoplasia. To the best of our knowledge, no human studies have addressed this question and there are only a handful of animal-based studies that have investigated it [148,151,152,153,154]. For instance, using the direct counting method, Nicks et al. (1989) concluded that the total number of fibers in the long head of the triceps brachii muscle of rats was not altered after eight weeks of forelimb immobilization [148]. Likewise, Templeton et al. (1988) found that four weeks of hindlimb suspension did not alter the number of fibers in the rat soleus [154]. Consistent with this finding, Cardenas et al. (1977) subjected the soleus muscle of rats to four weeks of immobilization in a shortened position and, after counting the number of fibers per mid-belly cross-section, concluded that the disuse did not alter the number of fibers per muscle [152]. Indeed, we are only aware of one study that has reported disuse-induced hypoplasia. Specifically, in this study, Booth and Kelso (1973) subjected the soleus muscle of rats to four weeks of immobilization in a neutral position [151]. The number of muscle fibers per cross-section was then counted and, in contrast to the work of Carenas et al. (1977), it was concluded that immobilization led to a 24% reduction in the number of fibers. Importantly, in this study, the region of the muscle via which the cross-sections were obtained was not specified, and this is important because, as highlighted in Figure 4B, taking a cross-section from the soleus at a region that is proximal or distal to the mid-belly could lead one to mistakenly conclude that there was disuse-induced hypoplasia. Thus, after considering this explanation, we feel fairly confident in concluding that disuse does not induce hypoplasia.

## 6. Disuse-Induced Atrophy at the Ultrastructural Level

Up to this point, we have concluded that disuse can induce both longitudinal and radial atrophy of muscle fibers. According to our basic guideline, these changes will be mediated via the alterations that occur at the preceding level within the hierarchy of the structural elements. In this case, the preceding level represents the ultrastructure of the muscle, and at this level, one will find that ~80% of the muscle fiber volume is composed of myofibrils [70,71]. Given that myofibrils make up the bulk of the muscle fiber volume, it can be reasoned that longitudinal and/or radial atrophy of the muscle fibers will be largely mediated via changes that occur at the level of the myofibrils. Accordingly, in this section, we will explore how disuse induces atrophy at the level of the myofibrils. 

### 6.1. Longitudinal Atrophy of Muscle Fibers

As summarized in the previous section, a compelling body of studies has shown that disuse in a shortened position can induce the longitudinal atrophy of the muscle fibers. Since muscle fibers are filled with myofibrils, and the myofibrils are composed of an in-series repetition of sarcomeres (Figure 1A), it follows that a decrease in the length of muscle fibers should be driven by a decrease in the length of the sarcomeres and/or the loss of in-series sarcomeres. When considering these two possibilities, it is important to recognize that if the decrease in fiber length were mediated via a decrease in sarcomere length, then the shortened sarcomeres would not be able to function at their optimal length for force production [82,158]. This point, coupled with the fact that the resting/optimal length of sarcomeres (2.0–2.5 µm) is highly conserved within a given species [72,73], suggests that the muscle fibers will be faced with a strong drive to restore the resting/optimal length of its sarcomeres. To accomplish this, the muscle fibers would have to eliminate in-series sarcomeres, and, not surprisingly, this is what has been observed in numerous studies [41,82,84,126,127,129,149,159,160]. For instance, Shah et al. (2001) reported that 28 days of immobilizing the mouse soleus muscle in a shortened position led to a 26% decrease in fiber length, and this was accompanied by a 26% decrease in the number of in-series sarcomeres [129]. Likewise, Tabary et al. (1972) found that 21–43 days of immobilizing the soleus muscle of cats in a shortened position led to an ~34% decrease in fiber length and that this was associated with an ~40% reduction in the number of in-series sarcomeres [126]. In addition to immobilization, similar decrements in both fiber length and the number of in-series sarcomeres have also been reported to occur in response to hindlimb suspension [41,127]. Thus, it can be strongly concluded that when muscles are subjected to disuse in a shortened position, it will lead to longitudinal atrophy of the muscle fibers, and this effect is largely mediated via the elimination of in-series sarcomeres. With that being said, it bears noting that all of the aforementioned studies were focused on muscles that were subjected to disuse in a shortened position, and whether the same phenomenon happens when muscles are subjected to disuse in a neutral position has, to the best of our knowledge, not been addressed.

Having established that disuse can lead to the elimination of in-series sarcomeres, one is faced with the question of how this occurs. Unfortunately, to date, the mechanism(s) that drive this phenomenon remain far from defined. Nonetheless, multiple studies have suggested that the elimination of in-series sarcomeres might occur in regions of myofibrillar disruption that are generally referred to as (i) central core-like lesions (CCLs) and (ii) segmental necrosis. 

CCLs are the most widely reported type of myofibrillar disruption [42,43,47,54,146,161,162,163,164,165,166,167,168], and conventional CCLs can be described as centrally located circular disruptions of the myofibrils within a cross-section of a muscle fiber (Figure 5A). However, less widely appreciated patches of irregularly shaped CCL-like disruptions can also be found throughout the entirety of muscle fibers that are undergoing disuse-induced atrophy (Figure 5A–C) [42,43,168]. In both cases, a close examination of these regions reveals that they are enriched with signs of active remodeling, including a disarray of the myofilaments, loss of mitochondria, and fragmentation of Z-lines [42,43,54,161,163]. Interestingly, a large number of studies have reported that CCLs are very common in muscles that have been subjected to disuse in a shortened position [42,43,47,54,146,164,165,166,167,168]. However, such regions are rarely found in muscles that have been subjected to immobilization in a lengthened position [47,54,165]. This is an important point because when muscles are subjected to immobilization in a lengthened position, they will generally still undergo atrophy, but the atrophic response will not involve the in-series elimination of sarcomeres [81,126,129,159,160]. Thus, the presence of CCLs appears to occur specifically in response to atrophic conditions that lead to the in-series elimination of sarcomeres. 

The aforementioned points are all consistent with the notion that the elimination of in-series sarcomeres occurs at CCLs. However, numerous studies have also reported that CCLs are largely confined to Type I fibers [43,47,54,146,164,165,166]. For example, Riley et al. (1990) subjected the soleus muscle of rats to 13 days of hindlimb suspension and found that ~92% of the fibers with CCLs were Type I fibers [43]. To the best of our knowledge, there is no evidence, or reason to think, that the loss of in-series sarcomeres would be confined to type I fibers. Thus, although intriguing, the notion that the elimination of in-series sarcomeres occurs at CCLs remains highly questionable and additional studies will be needed to resolve this issue. Moreover, although conventional CCLs and patches of CCL-like disruptions both show signs of active remodeling, they might be involved in distinct biological processes. Indeed, Murakami et al. (2008) reported that irregularly shaped areas of ‘sarcomeric disarray’ (i.e., patches of CCL-like disruptions) were readily apparent in rat soleus muscles after 7 days of hindlimb suspension, but the presence of conventional CCLs only became apparent after 14 days of hindlimb suspension [168]. The different temporal nature of their appearance suggests that conventional CCLs and patches of CCL-like disruptions could represent biologically distinct areas of remodeling, and thus, future studies should take care to classify these areas as different types of myofibrillar disruptions. Consistent with the nomenclature of Murakami et al. (2008), we propose that centrally located circular disruptions should be classified as conventional CCLs, and the irregularly shaped CCL-like disruptions should be referred to as areas of sarcomeric disarray (SDA). 

The other type of myofibrillar disruption that has been implicated in the in-series elimination of sarcomeres is called segmental necrosis [42,43,163,167,169,170]. Unlike CCLs, segmental necrosis typically presents as a destruction of an entire segment of the muscle fiber and is associated with the infiltration of mononuclear cells and neutrophils as well as signs of elevated phagocytic activity [163,169,170]. Two examples of segmental necrosis are shown in Figure 5D,E, and it has been proposed that these regions could serve as areas in which large segments of in-series sarcomere are eliminated [167,169]. Importantly, just like CCLs, areas of segmental necrosis are generally confined to muscles that have been subjected to disuse in a shortened position, but their appearance is quite rare when compared with that of CCLs. For instance, Riley et al. (1990) reported that only ~3% of rat soleus muscle fibers possessed regions of segmental necrosis after 10 days of hindlimb suspension, whereas ~30% of the fibers showed signs of CCLs after 13 days [43]. Undoubtedly, there is still much to learn about the function of CCLs, areas of segmental necrosis, and the mechanisms that give rise to them. The only thing that we can confidently conclude at this point is that additional studies will be needed to determine whether these are truly the sites in which in-series sarcomeres are eliminated during the longitudinal atrophy of muscle fibers. 

### 6.2. Radial Atrophy of Muscle Fibers

In Section 5.2, we reviewed the evidence which indicates that radial atrophy of the muscle fibers is a major contributor to disuse-induced atrophy. Since the bulk of the muscle fiber volume is composed of myofibrils, it can be reasoned that radial atrophy of the muscle fibers will primarily be mediated via (i) a decrease in the CSA of the existing myofibrils (i.e., radial atrophy of the myofibrils) and/or (ii) a decrease in the number of myofibrils per muscle fiber (i.e., myofibril hypoplasia) (Figure 6). Importantly, loss of the intermyofibrillar components (e.g., sarcoplasmic reticulum and mitochondria) could also contribute to the radial atrophy of the muscle, but we are not aware of any studies that have addressed this possibility. Therefore, in this section, we will review the studies that have addressed whether the disuse-induced radial atrophy of muscle fibers is mediated via radial atrophy of the myofibrils and/or myofibril hypoplasia. 

#### 6.2.1. Radial Atrophy of Myofibrils

Radial atrophy of myofibrils is defined as a decrease in the CSA of the myofibrils. Having illustrated how radial atrophy of the myofibrils can lead to radial atrophy of the muscle fiber (Figure 6), we will now consider the studies that have investigated whether disuse leads to changes in myofibril CSA. Sadly, we could not find any studies that have directly measured myofibril CSA; however, there are at least two studies that reported measurements of myofibril diameter, and in both cases, it was concluded that disuse leads to a decrease in myofibril diameter [163,171]. For instance, Riley et al. (1990) reported a 23% decrease in the diameter of the myofibrils of the adductor longus muscles of rats after 12.5 days of space flight [163]. Likewise, Giordano et al. (2014) reported a 50% reduction in the diameter of myofibrils in the plantaris muscle of rats that had been subjected to 10 days of hindlimb suspension [171]. Consistent with these results, several qualitative studies have also reported that disuse leads to a reduction in myofibril diameter [43,172,173,174,175]. Hence, the available data suggest that radial atrophy of muscle fibers is mediated, at least in part, via radial atrophy of the myofibrils. 

#### 6.2.2. Myofibril Hypoplasia

Myofibril hypoplasia refers to the loss of myofibrils, and as illustrated in Figure 6, myofibril hypoplasia could also lead to the radial atrophy of the muscle fibers. However, we are not aware of any studies that have directly addressed this topic. Nevertheless, there is one study that, after some extrapolation, suggests that the radial atrophy of muscle fibers that occurs during space flight is not mediated via myofibril hypoplasia [163]. Specifically, Riley et al. (1990) reported that 12.5 days of space flight led to a 36% decrease in the CSA of the muscle fibers and that was accompanied by a 23% decrease in myofibril diameter. If myofibrils are viewed as round cylinders, then a 23% decrease in the myofibril diameter would be equivalent to a 41% decrease in the myofibril CSA. Hence, at least in this study, the decrease in muscle fiber CSA could be fully explained by the radial atrophy of the myofibrils, and therefore suggests that disuse did not lead to myofibril hypoplasia. Despite this point, there is a major gap in our knowledge about whether disuse leads to myofibril hypoplasia, and this is certainly a topic that is worthy of further investigation.

#### 6.2.3. Mechanisms That Might Promote the Radial Atrophy of Myofibrils

As described above, current evidence indicates that the radial atrophy of muscle fibers during disuse is mediated, at least in part, via radial atrophy of the myofibrils. Since myofibrils are largely composed of thick and thin myofilaments, it can be inferred that radial atrophy of the myofibrils will be primarily mediated via the loss of these myofilaments. Importantly, the number of myofilaments per myofibril depends on the CSA of the myofibril and the density of the myofilaments (i.e., the number of myofilaments per CSA) [176,177]. Thus, if the CSA of myofibril becomes smaller and the density of the myofilaments does not increase, then the number of myofilaments must decrease. Fortunately, myofilament density has been investigated in several models of disuse atrophy, and none of these studies observed an increase in the density of either the thick or thin myofilaments [43,172,173,174,175,178]. Accordingly, it can be strongly concluded that the disuse-induced radial atrophy of myofibrils is associated with the loss of myofilaments. 

In the simplest terms, a disuse-induced loss of myofilaments will occur when there is a negative balance between the rate of protein synthesis and the rate of protein degradation, and a plethora of human and animal studies have shown that disuse leads to a decrease in the rate of protein synthesis [53,179,180,181,182,183,184,185,186,187,188,189,190,191,192,193,194,195,196,197,198,199]. For instance, nearly 50 years ago, Goldspink reported that immobilization of the rat soleus muscle in a shortened position led to a rapid (within 6 h) and sustained (>7 days) decrease in the rate of protein synthesis [179], and a multitude of similar observation have been reported when muscles are subjected to disuse in a neutral position. As a case in point, Gibson et al. (1987) reported that 5 weeks of immobilizing the quadriceps of humans in a neutral position led to a 26% decrease in the rate of protein synthesis [183]. Likewise, De Boer et al. (2007) found that 10 days of unilateral lower limb suspension led to a 53% decrease in the rate of myofibrillar protein synthesis in the quadriceps [189].

Although a clear and consistent body of evidence indicates that disuse leads to a decrease in the rate of protein synthesis, whether disuse alters the rate of protein degradation is questionable. For instance, some human and animal studies have revealed that disuse is associated with an increase in the rate of protein degradation [179,181,200,201], and numerous studies have found that disuse leads to an increase in the expression of genes that regulate the ubiquitin–proteasome pathway of protein degradation [106,202,203,204,205,206,207]. However, several studies have also shown that disuse does not alter the rate of protein degradation [185,192,208,209]. Given these contradictory results, we believe that the notion that disuse alters the rate of protein degradation should be viewed as a subject of debate [210,211,212]. 

Based on the studies described above, it seems fair to conclude that the disuse-induced loss of myofilaments is mediated, at least in part, via a decrease in the rate of protein synthesis. Given this conclusion, one is then faced with the question of where new myofilaments are synthesized and deposited within the myofibrils, as the answer to this question would provide insight into how disuse leads to a loss in the total number of myofilaments within the myofibril. New myofilaments are synthesized by ribosomes, and previous studies have reported that ribosomes are particularly prominent in the intermyofibrillar space, as well as in the subsarcolemmal and perinuclear regions of the fiber [213,214,215]. Importantly, these same studies have also reported that ribosomes are not found within the intramyofibrillar space. Thus, according to the positions of the ribosomes, one would expect new myofilaments to be synthesized and initially deposited at the periphery of the myofibrils. Consistent with this proposition, Dr. Morkin (1970) used ^3^H-leucine to label newly synthesized proteins and then used electron microscope-based autoradiography to identify where the newly synthesized proteins accumulated, and the results indicated that the deposition was primarily confined to the periphery of the myofibrils [216]. As such, it would appear that the radial atrophy of myofibrils that occurs during disuse is due, at least in part, to a decrease in the rate at which new myofilaments are synthesized and deposited at the periphery of myofibrils.

In addition to the above-mentioned conclusion, it also bears noting that a handful of studies have reported that muscles subjected to disuse have myofibrils that exhibit a moth-eaten appearance [43,172,174,178,217,218]. For instance, as shown in Figure 7, Riley et al. (1987) reported that 7 days of spaceflight led to an increase in the prevalence of gaps in the myofilaments around the periphery of the myofibrils, as well as an increase in the focal loss of myofilaments within the myofibrils [174]. The same type of peripheral and intramyofibrillar moth-eaten appearance has also been reported by others [43,178]. Hence, when taken together, these observations provide support for the notion that radial atrophy of myofibrils is not simply mediated via decreased myofilament deposition at the periphery of myofibrils, but also via the focal loss of myofilaments. 

Having established that radial atrophy of myofibrils is associated with the focal loss of myofilaments, one is faced with questions about the mechanism that drives this phenomenon. To the best of our knowledge, no studies have addressed this topic, but we suspect that it may be attributed to a sub-group of myofilaments that have been referred to as easily releasable myofilaments (ERMs). Specifically, ERMs are a sub-group of myofilaments that can be easily released from the myofibrillar fraction of a muscle lysate via gentle agitation with an ATP-containing solution [219,220,221,222,223]. It is thought that the presence of ATP dissociates the actin–myosin interaction and facilitates the release of myofilaments that are weakly associated with the Z disc. For instance, Belcastro et al. (1991) reported that only ~2–4% of the myofilaments are released from rabbit myofibrils when they are agitated with a solution that does not contain ATP, and this proportion increases to ~28% when ATP is added to the agitation solution [222]. It has also been reported that ERMs contain the major constituents of the myofibrils ‘myosin heavy chain and actin’, but very little alpha-actinin and desmin [221,222,223], suggesting that the contractile myofilaments are more susceptible to release in an ATP-containing solution than the Z-line components and intermediate myofilaments. It also bears noting that previous studies have reported that the proportion of ERMs in muscle lysates increases during atrophic conditions such as food deprivation and sepsis [224,225,226], as well as when the myofibrils are pretreated with proteases [221]. Thus, the potential for ERMs to play a role in the disuse-induced loss of myofilaments is intriguing and seems worthy of further investigation. 

## 7. Closing Remarks

As illustrated throughout this review, disuse can lead to a profound loss of skeletal muscle mass, and in Table 1 we have summarized the structural adaptations that are thought to mediate this response. In this table, we also list our opinion about the level of evidence that supports each of these adaptations, as well as some of the major gaps in knowledge that still exist. After reading this review we hope that the reader can appreciate that our knowledge about the structural adaptations that mediate disuse atrophy is far from complete. Indeed, a number of foundationally important questions have yet to be answered and, in many instances, we believe that the answers to these questions will be essential for the advancement of the field. 

## Figures and Tables

**Figure 1 cells-12-02811-f001:**
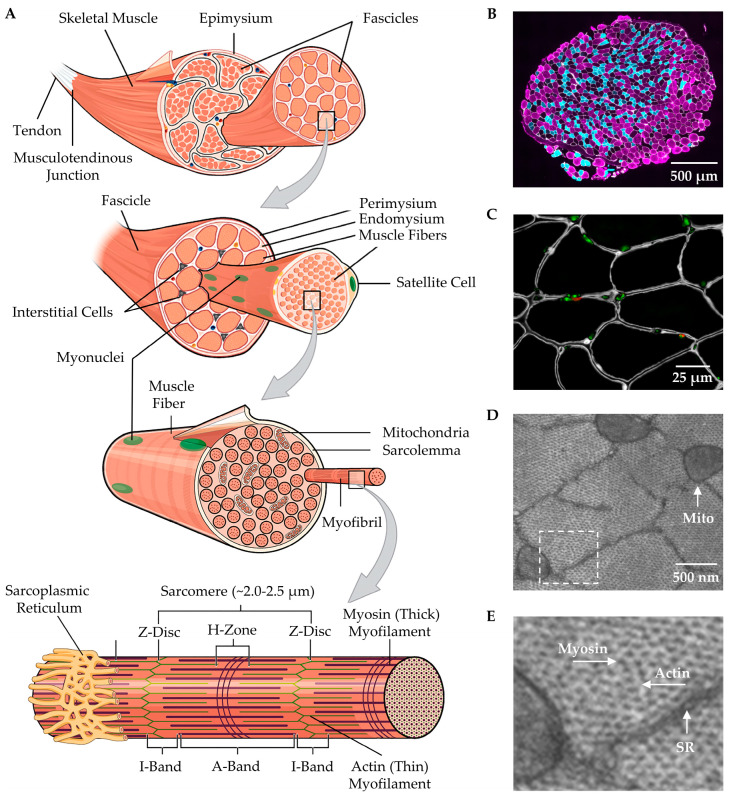
Overview of skeletal muscle structure. (**A**) Illustration of major structural components of skeletal muscle. (**B**) Immunohistochemical staining (IHC) of a mouse plantaris muscle cross-section for laminin (white) as well as Type IIA (cyan) and Type IIB muscle fibers (magenta) were used to highlight the endomysium that surrounds individual muscle fibers. (**C**) IHC of a tibialis anterior muscle cross-section was used to identify a major component of the endomysium (laminin, white), a marker of satellite cells (Pax7, red), and nuclei (green). (**D**,**E**) Electron microscopy of mouse plantaris muscle cross-section was used to highlight the sarcoplasmic reticulum (SR) and mitochondria (mito) that surround individual myofibrils. (**E**) Higher magnification of the boxed region in (**D**) reveals the presence of the myosin (thick) and actin (thin) myofilaments. The figure is adapted from Jorgenson et al., 2020 [32].

**Figure 2 cells-12-02811-f002:**
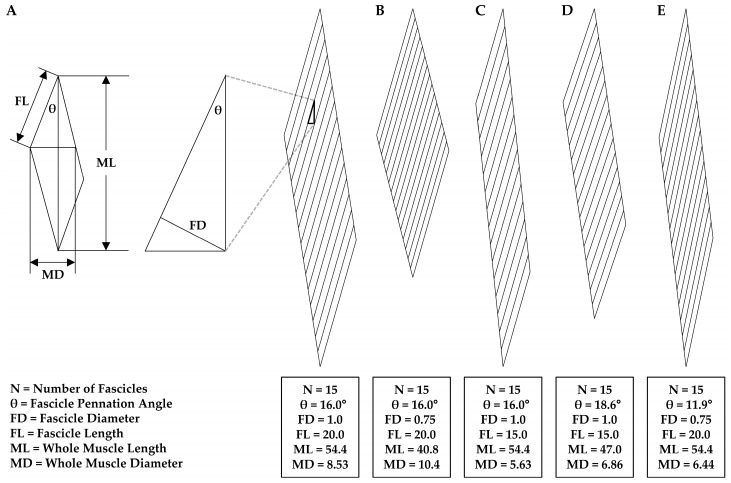
Illustration of how longitudinal and radial atrophy of fascicles could influence the structural adaptations observed at the whole muscle level. (**A**) Description of a hypothetical muscle and its basic architectural properties. The geometric model of Maxwell et al. (1974) [108] was used to predict how only a 25% decrease in fascicle diameter (**B**), or only a 25% decrease in fascicle length (**C**), would impact measurements of whole muscle length and whole muscle diameter. (**D**) Prediction of how a 25% decrease in fascicle length plus a 14% decrease in whole muscle length would impact the whole muscle diameter and pennation angle of the fascicles. (**E**) Prediction of how a 25% decrease in fascicle diameter would impact the whole muscle diameter and pennation angle of the fascicles when whole muscle length is fixed.

**Figure 3 cells-12-02811-f003:**
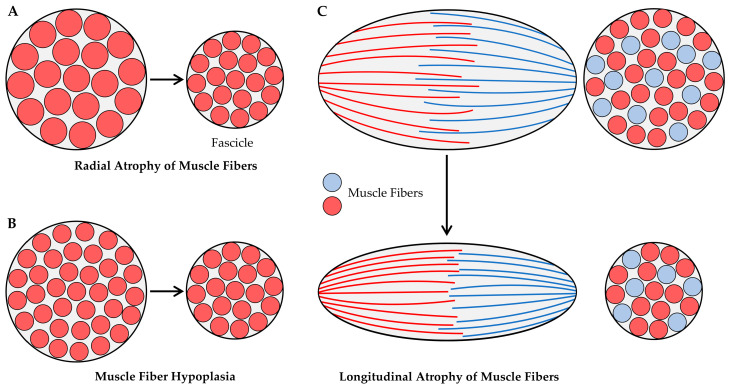
Illustration of the mechanisms that could lead to radial atrophy of fascicles. (**A**) Radial atrophy of muscle fibers, (**B**) muscle fiber hypoplasia, and (**C**) longitudinal atrophy of muscle fibers that exhibit intrafascicular terminations such as those found in the sartorius and gracilis muscles of humans [120,124]. The figure is adapted from Jorgenson et al., 2020 [32].

**Figure 4 cells-12-02811-f004:**
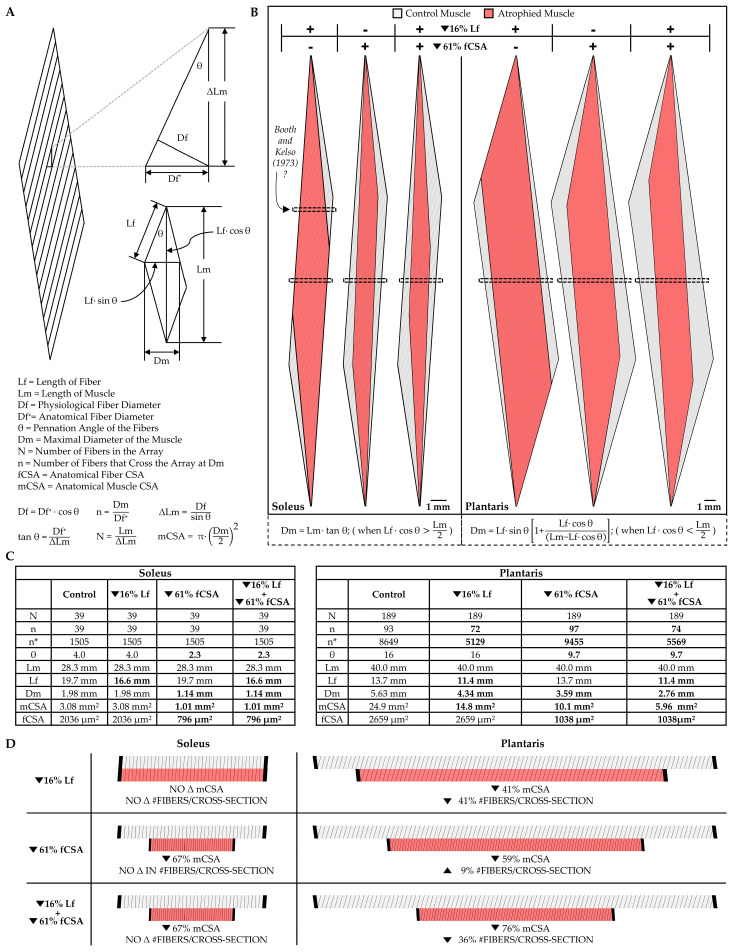
Illustration of how changes in fiber length and fiber CSA can alter the number of muscle fibers that appear in a mid-belly cross-section. (**A**) Key formulas from the geometric model that was used to predict the architectural properties of skeletal muscle [108]. (**B**) Illustration of how a 16% decrease in the fiber length and/or a 61% decrease in the fiber CSA would exhibit different effects on the number of muscle fibers that appear in a mid-belly cross-section of the soleus and the plantaris muscles. Note: these are high-resolution images, and details at the single fiber level should be visible when zooming in. Also note that the potential site of cross-section by Booth and Kelso (1973) has been highlighted [151]. (**C**) Quantitative results from the illustrations in (**B**), where the data for the control muscles was derived from previously reported architectural properties [84,127,128,143,155,156,157], and the rationale for the 16% decrease in the fiber length and the 61% decrease in the fiber CSA being derived from the average decrease in these values that were reported in the following studies [81,82,84,127,128,143]. (**D**) Illustration of how a 16% decrease in fiber length and/or a 61% decrease in fiber CSA have no impact on the number of fibers that appear in mid-belly cross-sections (hatched boxes in (**B**)) of the soleus muscle, but the same changes in fiber length and/or fiber CSA dramatically alter the number of fibers that appear in mid-belly cross-sections of the plantaris.

**Figure 5 cells-12-02811-f005:**
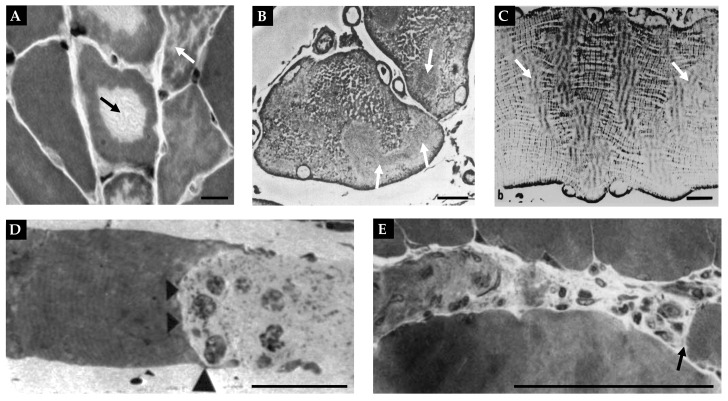
Representations of CCLs and segmental necrosis. (**A**) Cross-section of soleus muscle fibers from rats subjected to 12 days of hindlimb suspension. The black arrow points to a conventional CCL, and the white arrow points to a fiber with patches of CCL-like disruptions; copied with permission from [165]. (**B**,**C**) Cross-sections and longitudinal-sections of the soleus muscle of a rabbit subjected to 7 days of hindlimb suspension, respectively. The white arrows point to patches of CCL-like disruptions found throughout the muscle fiber; copied with permission from [42]. (**D**) Longitudinal-section of a soleus muscle fiber from a rat 1 day after being subjected to disuse via tenotomy. The black arrow heads point to a large portion of muscle fibers that has undergone segmental necrosis; copied with permission from [169]. (**E**) Longitudinal-section of soleus muscle fibers from rats subjected to 7 days of immobilization in a shortened position. The black arrow refers to an area of segmental necrosis near the muscle-tendon junction; copied with permission from [167]. Scale bar = 10 µm in all images.

**Figure 6 cells-12-02811-f006:**
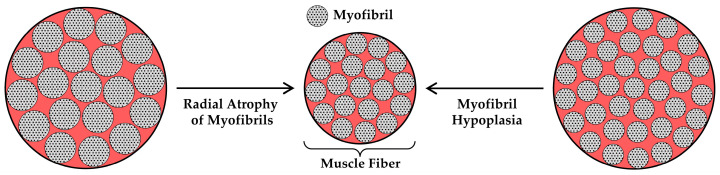
Illustration of how radial atrophy of myofibrils and/or myofibril hypoplasia can contribute to the radial atrophy of the muscle fibers. The figure is adapted from Jorgenson et al., 2020 [32].

**Figure 7 cells-12-02811-f007:**
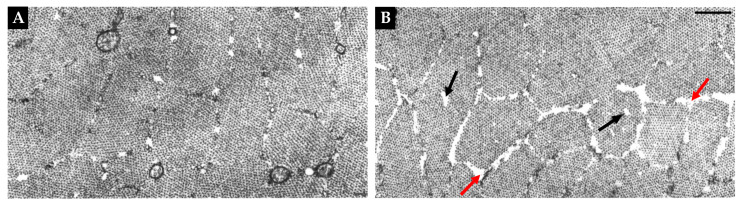
Representations of the moth-eaten appearance of myofibrils that occurs during disuse. Electron micrographs from rat soleus muscles that have been subjected to 7 days of a control condition (**A**), or 7 days of spaceflight (**B**). Red arrows point to the presence of gaps in the myofilaments around the periphery of the myofibrils. Black arrows point to the focal loss of myofilaments within the myofibrils. Scale bar = 0.5 µm; copied with permission from [174].

**Table 1 cells-12-02811-t001:** The Structural Adaptations that Mediate the Disuse-Induced Atrophy of Skeletal Muscle.

Adaptation	Evidence	Major Gaps in Knowledge
**Longitudinal Atrophy of Fascicles**	High (*for disuse in a shortened position*)	Does longitudinal atrophy of fascicles happen when muscles are subjected to disuse in a neutral position?
**Radial Atrophy of** **Fascicles**	High	To what extent (if any) does muscle fiber hypoplasia contribute to the radial atrophy of fascicles? To what extent does longitudinal atrophy of the muscle fibers contribute to the radial atrophy of fascicles?
**Longitudinal Atrophy of Muscle Fibers**	High (*for disuse in a shortened position*)	Does disuse in a neutral position induce longitudinal atrophy of muscle fibers? How are in-series sarcomeres eliminated?
**Radial Atrophy of** **Muscle Fibers**	Extremely High	To what extent is the radial atrophy of muscle fibers mediated via the radial atrophy of myofibrils vs. myofibril hypoplasia? Does the disuse-induced loss of the interstitial elements make a substantive contribution to the radial atrophy of muscle fibers?
**Radial Atrophy of** **Myofibrils**	Moderate	How are myofilaments eliminated during disuse?
**Myofibril Hypoplasia**	Very Low	If disuse leads to the myofibril hypoplasia, then how are the myofibrils eliminated?

**Note:** The underlined major gaps in knowledge are considered to be essential for the advancement of the field.

## Data Availability

Not Applicable.

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
