# Peer review of "The Structural Adaptations That Mediate Disuse-Induced Atrophy of Skeletal Muscle"

_cells, 2023, doi:10.3390/cells12242811_

Round 1
Reviewer 1 Report
Comments and Suggestions for Authors
The review paper titled “the structural adaptations that mediate disuse-induced atrophy of skeletal muscle” provides a fantastic up-to-date recap of the current state of the literature. The authors identify some important key knowledge gaps and summarize important steps to fill these. The quality of the writing is exceptional, and is a pleasure to read. I provided some points below which you may consider in revising your review.
I found the definition of “disuse” to be non-intuitive. I get why you chose this, but a muscle casted at a long muscle length (with increased passive tension) is certainly a model of disuse (will add serial sarcomere, but lose parallel), however under your definition it would not be classified as such simply because there is passive tension – this has large reaching implications on the readers interpretation of your findings. The authors could have a section acknowledging that immobilization in a lengthened position often results in the loss of muscle mass (as shown by Hinks et al. Experimental), even though there is usually an increase in serial sarcomere number. In that case, there could be radial atrophy occurring simultaneously alongside longitudinal growth.
Regarding Denervation, here as been some work by Russ Hepple and others noting that muscle fibres before much more highly angular upon denervation, and become co-expressive in terms of fibre type. This may (or may not) fit into your ‘structural adaptations’ theme.
Well done
Author Response
Comments and Suggestions for Authors
The review paper titled “the structural adaptations that mediate disuse-induced atrophy of skeletal muscle” provides a fantastic up-to-date recap of the current state of the literature. The authors identify some important key knowledge gaps and summarize important steps to fill these. The quality of the writing is exceptional, and is a pleasure to read. I provided some points below which you may consider in revising your review.
- Thank you for this positive comment.
I found the definition of “disuse” to be non-intuitive. I get why you chose this, but a muscle casted at a long muscle length (with increased passive tension) is certainly a model of disuse (will add serial sarcomere, but lose parallel), however under your definition it would not be classified as such simply because there is passive tension – this has large reaching implications on the readers interpretation of your findings. The authors could have a section acknowledging that immobilization in a lengthened position often results in the loss of muscle mass (as shown by Hinks et al. Experimental), even though there is usually an increase in serial sarcomere number. In that case, there could be radial atrophy occurring simultaneously alongside longitudinal growth.
- We appreciate the comment concerning the disuse definition and, as stressed in the review, we were very careful in our development of this definition. In the review, we also acknowledge that our definition will exclude certain skeletal muscles that are often examined in commonly used models of disuse atrophy. For instance, during immobilization in a lengthened position, the dorsiflexor muscles experience increased passive tension and typically adapt with an increase in the number of in-series sarcomeres. According to our definition, disuse requires a maintenance or reduction in the mean passive tension as well as a reduction in the mean active tension generated by the muscle. Hence, the muscles that experience increased passive tension do not fulfill the definition of being subjected to disuse and, in our opinion, would be better referred to as being subjected to chronic stretch. With the use of our definition, there are no models of disuse that induce the in-series addition of sarcomeres and, to us, this makes perfect sense. On the other hand, describing disuse as something that can induce the in-series addition of sarcomeres would be counterintuitive. Hopefully, the reviewer can appreciate our position on this.
Regarding Denervation, here as been some work by Russ Hepple and others noting that muscle fibres before much more highly angular upon denervation, and become co-expressive in terms of fibre type. This may (or may not) fit into your ‘structural adaptations’ theme.
- As we clarified in Section 2, this review focuses only on the structural adaptations that mediate disuse atrophy in subjects that are free of any apparent disease. Denervation can be caused by a variety of different conditions, and numerous studies have also shown that denervation can lead to signs of muscle fiber death and a concomitant regenerative response (Yoshimura and Harii, Journal of Surgical Research, 1999 & Borisov and Carlson, 2000). Thus, the atrophy that occurs in response to denervation is likely the result of more than just disuse. Accordingly, studies that have employed denervation as a model of disuse and its associated atrophic responses have not been considered in this review
Well done.
- Thank you!
Reviewer 2 Report
Comments and Suggestions for Authors
The authors provide a clear review of the current understanding of mechanisms that contribute to muscle atrophy associated with disuse. The authors carefully separate the consideration of mechanisms at different level of muscle structure, including an interesting discussion of the consequences of decreases in fascicle length and/or diameter on atrophy of pennate muscles. The review includes discussion of limitations of previous work in the topic area and identifies topics for which more work is needed.
Suggestions:
1. Page 4, seventh last line: “are” should be changed to “is”.
2. Page 4, last sentence: “sliding filament theory of muscle contraction” should be changed to “length/tension relationship of muscle” or something equivalent. The authors should then consider whether references 76 and 77 are still appropriate for the edited sentence.
3. Page 6, fourth line of first full paragraph: “in” should be inserted after “maintained”.
4. Page 7, sixth last line of first full paragraph: “are” should be changed to “is” as “number” is singular even though it refers to multiple possibilities in this sentence.
5. Page 14, fifth line of second full paragraph: “of” should be inserted after “entirety”.
6. Page 14, second full paragraph: Goldspink and co-workers reported the addition of sarcomeres in stretched, immobilized muscle in the 1970’s and/or 1980’s. The authors should consider whether this should be briefly discussed.
7. Page 16, second line: “interstitial” refers to the space between cells, not within cells. The authors should use different wording, such as other intracellular organelles or components. Also, “etc.” should be deleted as the parenthetical phrase as it is referring to only examples (e.g., …) and is not a comprehensive list.
Comments on the Quality of English LanguageNo concerns.
Author Response
Comments and Suggestions for Authors
The authors provide a clear review of the current understanding of mechanisms that contribute to muscle atrophy associated with disuse. The authors carefully separate the consideration of mechanisms at different level of muscle structure, including an interesting discussion of the consequences of decreases in fascicle length and/or diameter on atrophy of pennate muscles. The review includes discussion of limitations of previous work in the topic area and identifies topics for which more work is needed.
- Thank you for this positive comment.
Suggestions:
- Page 4, seventh last line: “are” should be changed to “is”.
- The sentence reads as “The area closest to the Z-discs that are occupied solely by the thin myofilaments …..” Since the use of the word “Z-discs” is plural we believe that the use of “are” (as opposed to “is”) was appropriate.
- Page 4, last sentence: “sliding filament theory of muscle contraction” should be changed to “length/tension relationship of muscle” or something equivalent. The authors should then consider whether references 76 and 77 are still appropriate for the edited sentence.
- Thank you for this recommendation. In this paragraph, we describe the arrangement of thick and thin myofilaments within the myofibril. In addition, we refer to the overlap between the thick and thin myofilaments throughout the A-band. Hence, we believe that using the “sliding filament theory of muscle contraction” is more appropriate.
- Page 6, fourth line of first full paragraph: “in” should be inserted after “maintained”.
- Inserted.
- Page 7, sixth last line of first full paragraph: “are” should be changed to “is” as “number” is singular even though it refers to multiple possibilities in this sentence.
- Changed.
- Page 14, fifth line of second full paragraph: “of” should be inserted after “entirety”.
- Inserted.
- Page 14, second full paragraph: Goldspink and co-workers reported the addition of sarcomeres in stretched, immobilized muscle in the 1970’s and/or 1980’s. The authors should consider whether this should be briefly discussed.
- We appreciate the suggestion, however, as we carefully defined in Section 2, disuse requires a maintenance or reduction in the mean passive tension as well as a reduction in the mean active tension generated by the muscle. Thus, when muscles are subjected to immobilization in a stretchted position they would not fulfill our definition of being subjected to disus.
- Page 16, second line: “interstitial” refers to the space between cells, not within cells. The authors should use different wording, such as other intracellular organelles or components. Also, “etc.” should be deleted as the parenthetical phrase as it is referring to only examples (e.g., …) and is not a comprehensive list.
- Thank you. The words “interstitial elements” have been changed to “intermyofibrillar components”, and “etc.” has been deleted.
Reviewer 3 Report
Comments and Suggestions for Authors
Journal
Cells (ISSN 2073-4409)
Manuscript ID
Cells-2740548
Review of “The Structural Adaptations That Mediate Disuse-Induced Atrophy of Skeletal Muscle”
This is a timely review exploring known works on the microstructural changes that occur during disuse atrophy of skeletal muscle. This review is a thoughtful triste on the known theories and gaps in knowledge of muscle microstructural alterations that occur as a result of disuse.
The figures and tables are a necessary addition to illustrate the more complex and spatial concepts.
The use of line numbers would have been useful for the reviewers.
Would the authors consider the discussion of region-specific muscle atrophy during disuse?
This would be if atrophy occurs at different depths of the muscle or in more proximal or distal locations (the later is touched upon).
Also, a discussion of more myofiber type-specific changes could be insightful.
Section 2. Maybe directly state the authors seek to isolate the conditions whereby intrinsic factors of decreased contraction are the primary contributors to atrophy as opposed to other conditions whereby extrinsic factors have a greater contribution.
Interesting that the TA is excluded for experiencing passive tension because, like the soleus, it also experiences significant atrophy.
The authors should also consider conditions where the animal is growing since changes in bone growth would impact longitudinal atrophy. Only studies with appropriate controls should be considered when examining that outcome.
Figure 2 is well constructed and explained.
Muscle fiber length: Consider “Hindlimb suspension suppresses muscle growth and satellite cell proliferation” https://pubmed.ncbi.nlm.nih.gov/2600016/ This may be most relevant if the atrophy occurs during a period of rapid bone growth. If bone is stable in length, the type of atrophy may be different.
This may be relevant. https://www.nature.com/articles/srep20061
How is density defined? https://pubmed.ncbi.nlm.nih.gov/15469952/
Stiffness: https://pubmed.ncbi.nlm.nih.gov/7761257/
“Spector et al. (1982), the limb of the rats may have been immobilized with the muscles in a neutral position, but the muscles were likely still experiencing a chronic lengthening stimulus from the longitudinal growth of the bones” What if the bone length growth is attenuated by the disuse?
Another concern with counting the number of muscle fibers per mid-belly cross-section and assuming that this is reflective of the total fiber number is that fiber could be lost in processing. Also, direct determination of the exact mid-belly in small rodent muscles is difficult to find when processing in OCT. This method could lead to an underestimation.
Author Response
Comments and Suggestions for Authors
Review of “The Structural Adaptations That Mediate Disuse-Induced Atrophy of Skeletal Muscle”
This is a timely review exploring known works on the microstructural changes that occur during disuse atrophy of skeletal muscle. This review is a thoughtful triste on the known theories and gaps in knowledge of muscle microstructural alterations that occur as a result of disuse.
The figures and tables are a necessary addition to illustrate the more complex and spatial concepts.
- Thank you for these positive comments.
The use of line numbers would have been useful for the reviewers.
- We apologize for not including line numbers in our submitted manuscript. This issue has been considered in the revised version.
Would the authors consider the discussion of region-specific muscle atrophy during disuse? This would be if atrophy occurs at different depths of the muscle or in more proximal or distal locations (the later is touched upon).
- This is a great question, and one would think that it would have been widely studied. However, during our review of the literature, we did not come across any studies that have adequately addressed this topic. Nevertheless, we believe that it would be worthy of further investigation.
Also, a discussion of more myofiber type-specific changes could be insightful.
- We appreciate the reviewer's recommendation. Indeed, this was a point that we directly addressed when we wrote our initial drafts of the manuscript. In our initial drafts we concluded that “it appears that during the early stages of disuse, type II fibers are particularly susceptible to radial atrophy, but this fiber-type dependent pattern is lost during more extended periods of disuse”. However, our conclusion was based on limited data, and the discussion of the supporting studies and their associated nuances became quite tangential and severely disrupted the overall flow of the manuscript. As such, we decided to exclude this topic from the manuscript and we would prefer to stick with that decision.
Section 2. Maybe directly state the authors seek to isolate the conditions whereby intrinsic factors of decreased contraction are the primary contributors to atrophy as opposed to other conditions whereby extrinsic factors have a greater contribution.
- We are very glad that the reviewer got the point of why we focused only here on the structural adaptations that mediate disuse atrophy in subjects that are free of any apparent disease. Extrinsic factors (such as malnutrition, denervation, age, genetics, certain medical conditions, etc.) may induce an increase in circulating atrophic factors and, in turn, lead to structural adaptations that are different from those in healthy individuals (see lines 58-64). Accordingly, only studies where intrinsic factors of decreased contraction are the primary contributors to disuse atrophy have been considered in this review.
Interesting that the TA is excluded for experiencing passive tension because, like the soleus, it also experiences significant atrophy.
- We have tried here to provide a more precise definition for disuse although it excludes certain skeletal muscles that are often examined in commonly used models of disuse atrophy (see lines 56-60). During immobilization in a lengthened position, the dorsiflexor muscles (such as TA and EDL) experience increased passive tension and show an increased in-series sarcomeres number. According to our definition, disuse requires a maintenance or reduction in the mean passive tension as well as a reduction in the mean active tension generated by the muscle. Hence, the muscles that experience increased passive tension do not fulfill the definition of being subjected to disuse. Instead, we would argue that that when muscles are immobilized in a lengthened position they are subjected to a form of chronic stretch rather than disuse.
The authors should also consider conditions where the animal is growing since changes in bone growth would impact longitudinal atrophy. Only studies with appropriate controls should be considered when examining that outcome.
- Thank you for this valuable comment. Indeed, we are aware of the issue that immobilized muscles of young individuals are likely still experiencing a chronic lengthening stimulus from the longitudinal growth of the bones and, when relevant, this point was considered (e.g., see lines 400-405). However, whenever possible, we intentionally excluded the consideration of data from studies that involved the use of subjects that were likely still experiencing longitudinal growth of the bones.
Figure 2 is well constructed and explained.
- Thank you for this positive comment.
Muscle fiber length: Consider “Hindlimb suspension suppresses muscle growth and satellite cell proliferation” https://pubmed.ncbi.nlm.nih.gov/2600016/ This may be most relevant if the atrophy occurs during a period of rapid bone growth. If bone is stable in length, the type of atrophy may be different.
- As mentioned above, we intentionally tried to exclude the consideration of data from studies that involved the use of subjects who were likely still experiencing longitudinal growth of the bones.
This may be relevant. https://www.nature.com/articles/srep20061
- Thank you. We have included this paper as further support for the conclusion that hindlimb unloading induces radial atrophy of muscle fibers (line 440).
How is density defined? https://pubmed.ncbi.nlm.nih.gov/15469952/
- The definition of myofilament density has been included(see lines 686 – 688)
Stiffness: https://pubmed.ncbi.nlm.nih.gov/7761257/
- The reason that the reviewer has shared this reference with us was not clear.
“Spector et al. (1982), the limb of the rats may have been immobilized with the muscles in a neutral position, but the muscles were likely still experiencing a chronic lengthening stimulus from the longitudinal growth of the bones” What if the bone length growth is attenuated by the disuse?
- As mentioned above, whenever possible, we intentionally excluded the consideration of data from studies that involved the use of subjects that were likely still experiencing longitudinal growth of the bones. Instead, we only considered such studies when no other data on a given topic was available. A complete consideration of this complicating variable is beyond the scope of our current review.
Another concern with counting the number of muscle fibers per mid-belly cross-section and assuming that this is reflective of the total fiber number is that fiber could be lost in processing. Also, direct determination of the exact mid-belly in small rodent muscles is difficult to find when processing in OCT. This method could lead to an underestimation.
- Thank you for pointing this out, it further highlights that caution needs to be exercised when making conclusions that are based on the number of fibers per cross-section.